# Pneumococcal colonisation is an asymptomatic event in healthy adults using an experimental human colonisation model

**Ashleigh Trimble**[1‡], **Victoria Connor**[1,2‡], **Ryan E. Robinson**[1,2], **Daniella McLenaghan**[1,2], **Carole A. Hancock**[2], **Duolao Wang**[1], **Stephen B. Gordon**[1,3], **Daniela M. Ferreira**[1], **Angela D. Wright**[1,3‡], **Andrea M. Collins**[1,2‡]*

**1** Clinical Sciences Department, Liverpool Life Sciences Accelerator, Liverpool, England, United Kingdom, **2** Respiratory Research Group at the Royal, Royal Liverpool and Broadgreen University Hospital Trust, Liverpool, England, United Kingdom, **3** Comprehensive Local Research Network, Northwest Coast, Liverpool, England, United Kingdom

‡ AT and VC are joint first authors on this work. ADW and AMC are joint last authors on this work.
* andrea.collins@lstmed.ac.uk

**Data Availability Statement:** All relevant data are within the paper and its Supporting Information files.

## Abstract

### Introduction

Pneumococcal colonisation is regarded as a pre-requisite for developing pneumococcal disease. In children previous studies have reported pneumococcal colonisation to be a symptomatic event and described a relationship between symptom severity/frequency and colonisation density. The evidence for this in adults is lacking in the literature. This study uses the experimental human pneumococcal challenge (EHPC) model to explore whether pneumococcal colonisation is a symptomatic event in healthy adults.

### Methods

Healthy participants aged 18–50 were recruited and inoculated intra-nasally with either *Streptococcus pneumoniae* (serotypes 6B, 23F) or saline as a control. Respiratory viral swabs were obtained prior to inoculation. Nasal and non-nasal symptoms were then assessed using a modified Likert score between 1 (no symptoms) to 7 (cannot function). The rate of symptoms reported between the two groups was compared and a correlation analysis performed.

### Results

Data from 54 participants were analysed. 46 were inoculated with *S. pneumoniae* (29 with serotype 6B, 17 with serotype 23F) and 8 received saline (control). In total, 14 became experimentally colonised (30.4%), all of which were inoculated with serotype 6B. There was no statistically significant difference in nasal ($p = 0.45$) or non-nasal symptoms ($p = 0.28$) between the inoculation group and the control group. In those who were colonised there was no direct correlation between colonisation density and symptom severity. In the 22% (12/52) who were co-colonised, with pneumococcus and respiratory viruses, there was no statistical difference in either nasal or non-nasal symptoms (virus positive $p = 0.74$ and virus negative $p = 1.0$).

**Funding:** This work received support from the Bill and Melinda Gates Grand Challenges Exploration Programme II to DF, SG, descriptor number: 10.12. The funders did not play any role in the study design, data collection and analysis, decision to publish, or preparation of the manuscript.

**Competing interests:** The authors have declared that no competing interests exist.

## Conclusion

Pneumococcal colonisation using the EHPC model is asymptomatic in healthy adults, regardless of pneumococcal density or viral co-colonisation.

## Introduction

*Streptococcus pneumoniae* (pneumococcus, SPN) frequently colonises the human nasopharynx, with 40–95% of infants and 10–25% of adults being colonised at any one time[1]. Pneumococcal/SPN colonisation rates also vary with geographical location, genetics and socioeconomic background[2]. SPN colonisation is a dynamic process. Although multiple SPN serotypes can both simultaneously and sequentially colonise, one serotype is usually the predominant current coloniser[3]. In addition interspecies competition occurs between resident flora and potential colonisers including *S.pneumoniae*, *H.influenzae* and *S.aureus[4]*.

Colonisation of the nasopharynx is considered a pre-requisite for SPN infections including pneumonia, sepsis, meningitis and otitis media. However, most colonisation episodes will not lead to subsequent disease. Pneumococcal colonisation is also thought to be the predominant source of immunological boosting against SPN infection in both children and adults[5, 6].

SPN colonisation appears to be asymptomatic in murine models[7] and in adults, however the current data are limited[8]. Previous studies in children have demonstrated mild nasal symptoms following SPN colonisation however when adjusted for age this relationship was weak[9]. Other studies have reported a relationship between symptom severity, pneumococcal density and pneumococcal/viral co-colonisation in children[10]. Pneumococcal colonisation may cause nasal symptoms in two ways; the bacteria could induce host secretions and inflammatory responses or in co-colonised individuals (pneumococcus and virus) due to viral proliferation inducing rhinitis[9]. Some studies have concluded that the presence of respiratory viruses and/or other bacteria within the nasopharynx are the main cause of symptoms; this colonisation in turn increases the rate of pneumococcal colonisation[9].

The novel experimental pneumococcal challenge model (EHPC) model mimics natural pneumococcal colonisation in healthy human adults and has been used to effectively study mucosal immunity and as a platform to test the efficacy of pneumococcal vaccines in randomised control trials[11, 12]. We aimed to use the EHPC model to investigate if the process of nasopharyngeal pneumococcal colonisation leads to symptoms.

## Methods

We recruited non-smoking healthy participants aged 18–60 years old (self-selection) as part of a larger EHPC dose ranging study between 24th May 2012 to 23rd October 2012, with follow up until January 2013. A modified CONSORT flow diagram is shown in Fig 1. Specimen collection and sample processing were conducted in Liverpool, UK. All participants gave written, informed consent. Ethical permission was granted by local NHS Research and Ethics Committee (REC) (11/NW/0592 Liverpool-East, date 11/10/2011). This study was retrospectively registered on the ISRCTN database (ISRCTN85403723) as this was not a mandatory requirement at the time of recruitment. All ongoing and related trials for this intervention are now prospectively registered with ISRCTN. Exclusion criteria included natural pneumococcal colonisation at baseline, any chronic medical condition or regular medication (study participation could put the participant at increased risk of pneumococcal disease) and regular contact with an at-risk individual such as young children (study participation could put the at-risk individual at increased risk of pneumococcal disease).

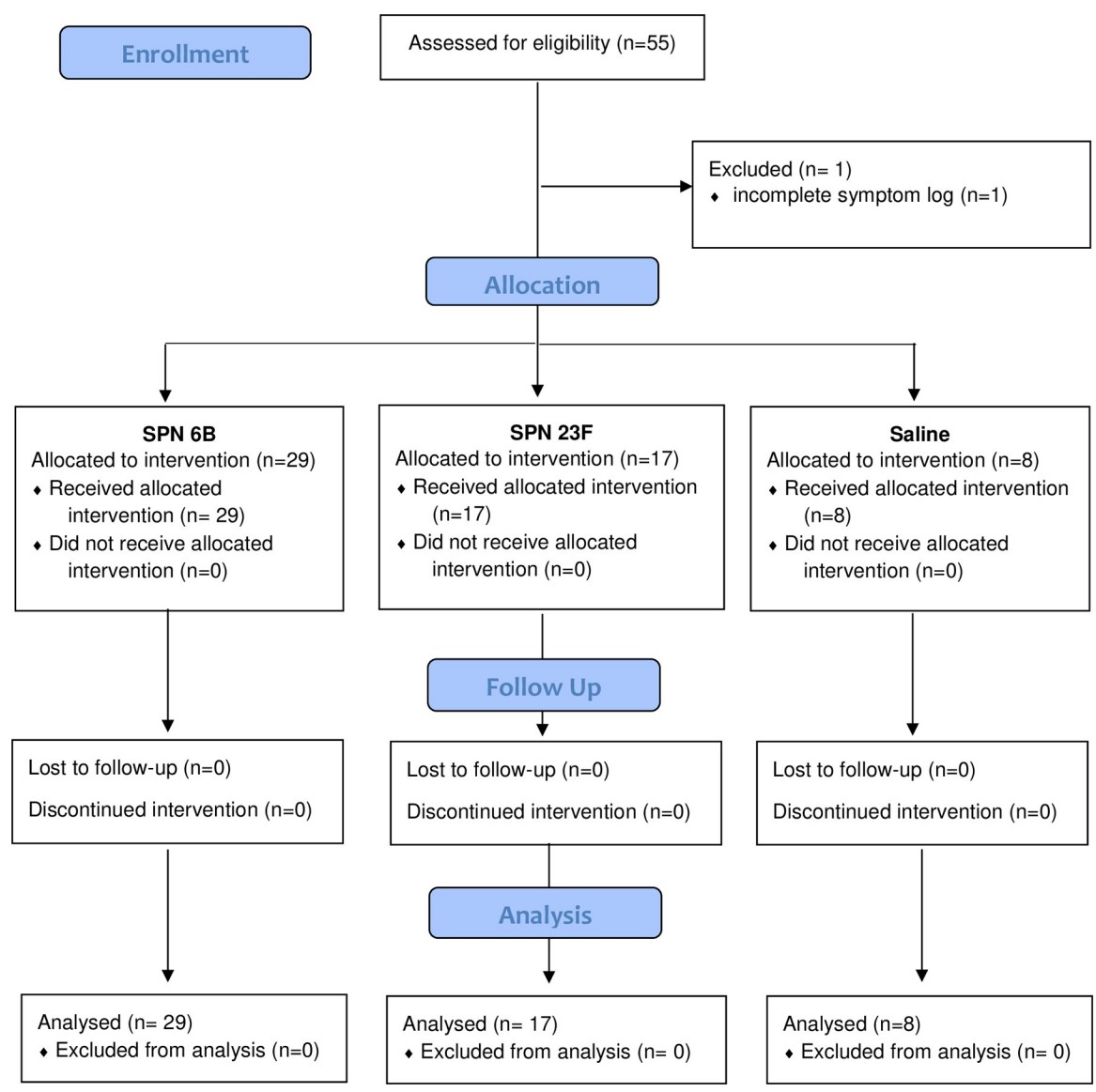

**Fig 1. Modified CONSORT diagram.**

Participants were nasally inoculated with $8x10^4$, $1.6x10^5$, or $3.2x10^5$ mid-log phase colony forming units (CFU) *S. pneumoniae* (prepared as previously described)[6]. Bacterial inoculation density was confirmed by serial dilutions of the inoculation stock onto blood agar (Oxoid). Two serotypes were used; 6B and 23F, both were fully sensitive to penicillin. 46 participants were allocated to be inoculated with *S. pneumoniae* (SPN 6B or 23F) as part of a dose-ranging study and 8 participants inoculated with saline as a control group. They were blinded to their groups.

Pre-inoculation oropharyngeal swabs were assayed for respiratory viruses using multiplex Polymerase Chain reaction (PCR) as previously published [13]. The PCR assay panel detected Influenza A and B, Respiratory syncytial virus, Human metapneumovirus, Human rhinovirus, Parainfluenza viruses 1–4 and Coronaviruses OC43, NL63, 229E and HKU1. Nasopharyngeal colonisation was assessed in nasal washes (Nacleiro technique, as previously described)

collected at day 2, 7 and 14 post inoculation[14]. Pneumococcal colonisation status and density in nasal washes was determined by classical culture as previously described[6, 14].

Participants were prompted to complete a daily symptom log on the day of inoculation (baseline) and daily for 7 days post-inoculation. The symptom log consisted of a 7-point visual analogue scale (a type of Likert scale) which assessed five nasal and five non-nasal symptoms [15]. The only modification to the validated questionnaire was the removal of 'mental function' as a non-nasal symptom (Fig 2). Scores ≥2 were considered 'symptomatic'. The score awarded at inoculation (day 0) was considered their baseline score, the participant was considered symptomatic if the score went above baseline and ≥ 2.

Graphical and statistical analyses were performed using GraphPad version 5.0 (GraphPad Software, La Jolla, CA, USA) and Microsoft Excel, with a p-value of <0.05 considered significant. Rates of symptoms reported between groups were compared using Fisher's exact tests and Chi square where appropriate. Correlation analysis was performed using Spearman's rank text. The daily symptom logs were collected at the next scheduled visit following completion.

## Results

Fifty-five participants were recruited with an age range of 19–49 years old over a 6- month period from24/04/2012-23/10/2012. Participants with incomplete symptom severity score logs

| Nasal Symptoms | | | | | | | |
|---|---|---|---|---|---|---|---|
| Sneezing | 1 | 2 | 3 | 4 | 5 | 6 | 7 |
| Runny nose | 1 | 2 | 3 | 4 | 5 | 6 | 7 |
| Congestion | 1 | 2 | 3 | 4 | 5 | 6 | 7 |
| Itchy nose | 1 | 2 | 3 | 4 | 5 | 6 | 7 |
| Postnasal drip | 1 | 2 | 3 | 4 | 5 | 6 | 7 |
| **Non-Nasal Symptoms** | | | | | | | |
| Eye symptoms | 1 | 2 | 3 | 4 | 5 | 6 | 7 |
| Throat symptoms | 1 | 2 | 3 | 4 | 5 | 6 | 7 |
| Cough | 1 | 2 | 3 | 4 | 5 | 6 | 7 |
| Ear symptoms | 1 | 2 | 3 | 4 | 5 | 6 | 7 |
| Headache | 1 | 2 | 3 | 4 | 5 | 6 | 7 |

| Severity score | |
|---|---|
| 1-2 | None to occasional limited episode |
| 3-4 | Mild to steady symptoms but easily tolerable |
| 5-6 | Moderately bothersome or symptoms hard to tolerate/may interfere with daily activities and/or sleep |
| 7 | Unbearably severe or symptoms are so bad/cannot function all of the time |

**A severity score <2 was considered asymptomatic.**

**Fig 2. Participant symptom log.** A severity score <2 was considered asymptomatic.

were excluded, therefore data from 54 participants were analysed. 46 participants were inoculated with SPN (29 with 6B, 17 with 23F) and 8 with saline (control group). Participants inoculated with 6B, 23F and saline were similar in age and gender distribution. In total, 14 participants became experimentally colonised (30.4%), all of which were inoculated with serotype 6B. None of the participants in the control group developed natural SPN colonisation during the study. Overall 72% (39/54) of participants reported either or both nasal or non-nasal symptoms during the 7 days post-inoculation. Of these symptoms, similar rates of nasal and non-nasal symptoms were reported; 59% (32/54) of participants reported nasal symptoms and 56% (30/54) reported non-nasal symptoms.

No statistical difference was seen between number of participants who reported symptoms in the experimental SPN positive or negative group. Similar rates of experimental SPN positive participants reported nasal symptoms (71%, 10/14) and non-nasal symptoms (57%, 8/14) compared to experimental SPN negative participants (50%, 16/32 in nasal (OR 2.50 [95% CI: 0.65–9.66] $P$ = 0.212) and non-nasal (OR 1.33 [95% CI: 0.38–4.73] $P$ = 0.754). See Fig 3.

Nasal SPN inoculation did not lead to greater rates of reported symptoms when compared to the control (saline inoculation) group, as show in Fig 4. Nasal symptoms were reported by 75% (6/8) of participants inoculated with saline compared to 57% (26/46) of those who were inoculated with SPN, no statistical difference was seen (p = 0.45). Similarly, no statistical difference was seen with the reporting of non-nasal symptoms 24/46 (52%) post-SPN inoculation compared to post-saline inoculation 6/8 (75%), (p = 0.28). Participants that reported 'any symptom' were higher in the control group 100% (8/8) compared to 67% (31/46) in the inoculation group, this was not statistically significant (p = 0.09).

Of the 14 participants colonised with SPN, colonisation density was measured at days 2 and 7. No direct correlation was seen between SPN density and the mean symptom severity score at day 2 and day 7 for nasal (p = 0.86 Spearman's correlation) and non-nasal symptoms (p = 0.83 Spearman's correlation), Fig 5.

Viral colonisation data was available for 96% (52/54) participants at baseline. Viral colonisation was detected in 22% (12/52) of participants, 2 were inoculated with saline and 10 with SPN [serotype 23F (n = 2) and 6B (n = 8)]. There was no increase in nasal or non-nasal symptoms respectively in virus positive 8/12 (67%) and 7/12 (58%) respectively compared to virus negative participants 23/40 (58% for both symptoms), p = 0.74 and p = 1.0. Experimental SPN colonisation rates were higher in the presence of virus 6/10 (60%) compared to 8/35 (23%) in virus negative participants (p = <0.05). Virus and SPN positive (Co-colonised) participants did not report greater rates of nasal or non-nasal symptoms [4/6, (60%) for both symptoms], when compared to SPN positive only [nasal symptoms 6/8 (75%), OR 0.67 [95% CI: 0.06–6.88], P = 1.000), non-nasal symptoms 4/8 (50%), OR 2.00 [95% CI: 0.22–17.90] P = 0.627] and virus positive only [nasal symptoms 3/4 (75%) OR 3.23 [0.30–35.13] $P$ = 0.600, non-nasal symptoms 2/4 (50%] OR 0.93 [95% CI: 0.11–7.59] $P$ = 1.000). This is shown in Fig 6.

## Discussion

This study, with a clear methodology, provides evidence supporting the hypothesis that experimental pneumococcal (SPN) colonisation in adults is an asymptomatic event. This novel use of a human challenge model allowed the study of pneumococcal colonisation and symptomology in a controlled environment. The carriage rate of pneumococcus in this study was 30.4% and all with SPN6B. This is higher than the 'natural rate', due to the artificial introduction of the bacteria directly into the nasopharynx. This SPN 6B (BHN418) is known from epidemiological studies to have a very low prevalence in the community, therefore the participants are unlikely to have been exposed to it previously. The carriage rate for those inoculated with 23F

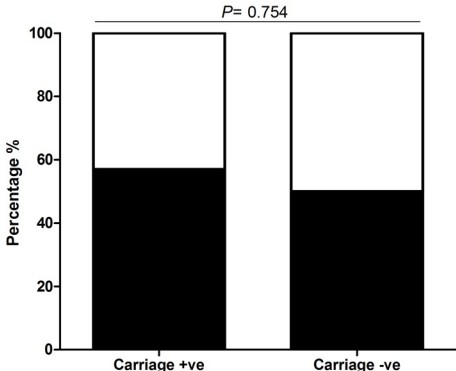

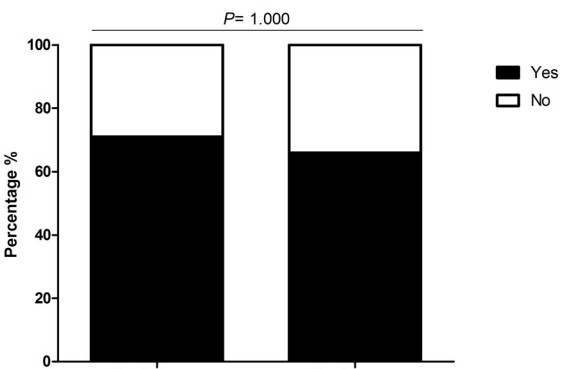

**Fig 3. Comparison of nasal, non-nasal and all symptoms between experimental SPN positive and negative participants.** Each bar chart shows the percentage of carriage positive (N = 14) and carriage negative (N = 32) participants, who reported symptoms (nasal, non-nasal and all symptoms) after inoculation with Streptococcus pneumoniae serotypes 6B or 23F. Participants were deemed symptomatic if they scored >2, or >1 point above baseline for any of the five nasal or non-nasal symptoms in the visual analogue scale. The number of participants reporting symptoms between carriage positive and negative status were statistically compared using Fishers Exact and deemed significant if P = ≤0.05. There was no significant difference in the number of participants reporting nasal (OR 2.50 [95% CI: 0.65–9.66] P = 0.212), non-nasal (OR 1.33 [95% CI: 0.38–4.73] P = 0.754) or all symptoms (OR 1.31 [95% CI: 0.33 to 5.16] P = 1.000) between carriage positive and carriage negative participants.

### Nasal Symptoms

*P*= 0.449

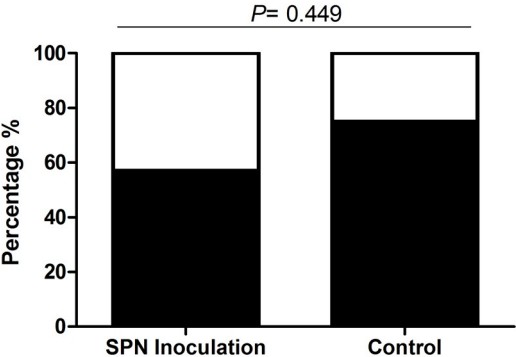

### Non-Nasal Symptoms

*P*= 0.277

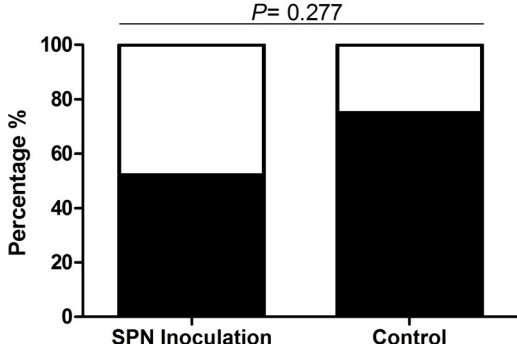

### All Symptoms

*P*= 0.089

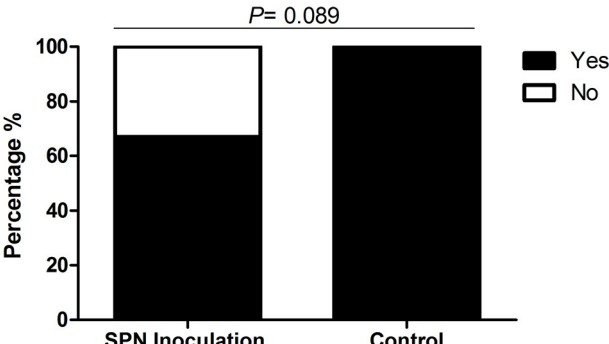

**Fig 4. Comparison of nasal, non-nasal and all symptoms between participants inoculated with S. pneumoniae compared to those inoculated with normal saline (control).** Each bar chart shows the percentage of participants inoculated with S. pneumoniae (SPN) serotypes 6B and 23F (N = 46) and normal saline (control) (N = 8) reporting symptoms (nasal, non- nasal and all symptoms) after inoculation. Participants were deemed symptomatic if they scored >2, or >1 point above baseline for any of the five nasal or non-nasal symptoms on the visual analogue scale. The number of participants reporting symptoms between inoculation with SPN and control were statistically compared using Fishers Exact and deemed significant if P = ≤0.05. There was no significant difference in the number of participants reporting nasal (OR 0.43 [95% CI: 0.08–2.38] P = 0.449), non-nasal (OR 0.36 [95% CI: 0.07–2.00] P = 0.277) and all symptoms (OR 0.12 [95% CI: 0.01–2.21] P = 0.089).

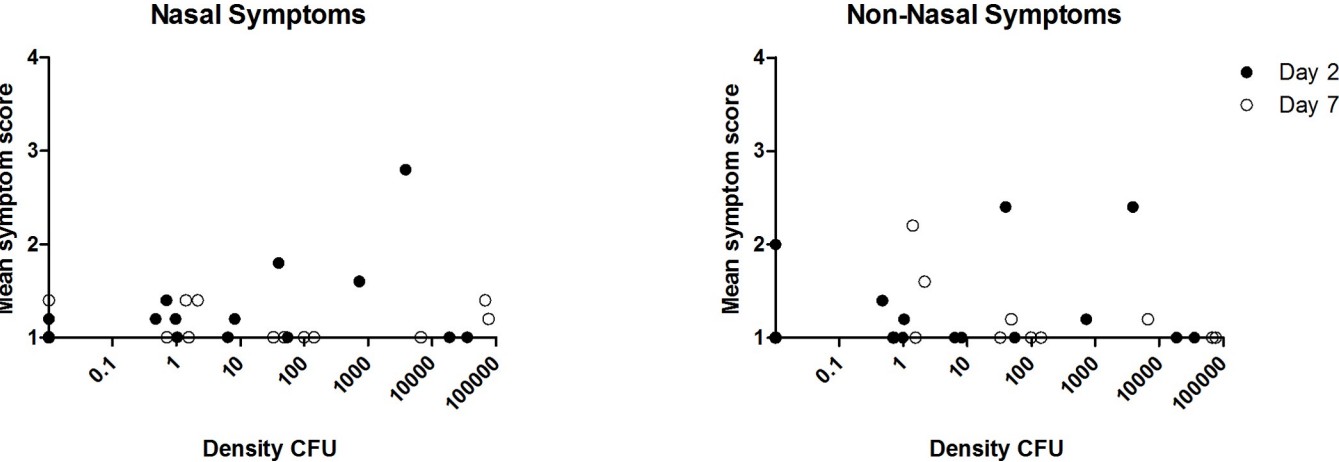

**Fig 5. Correlation between pneumococcal colonisation density (SPN positive participants) and mean nasal symptom severity scores at days 2 and 7.** Spearman's correlation was used to statistically analyse the correlation between bacterial colonisation density and the participants symptoms score on the visual analogue scale. Participants were deemed symptomatic if they scored >2, or >1 point above baseline for any of the five nasal or non-nasal symptoms in the visual analogue scale. No direct correlation was seen between SPN density and the mean symptom severity score at day 2 and day 7 for nasal (p = 0.86) or non-nasal symptoms (p = 0.83).

was 0%. The reasons for the variability in carriage rate between serotypes is unclear but thought to be related to the evasion of mucociliary clearance, host nutrient availability and niche competition[8].

The strengths of this study are the robust methodology used to assess symptom severity [15], the lack of recall bias (due to prospective daily data log completion) and the use of a control group. Using this novel human challenge model, the exact day of pneumococcal inoculation and the onset and termination of each SPN colonisation episode was known allowing association between symptoms and pneumococcal presence and density over time. The main limitations of our study was the total sample size (n = 54), the lack of randomisation for group allocation and the use of a single serotype of SPN.

Although a previous study in adults used a small sample size (n = 14) and did not include the methods used to support this conclusion[16], it agrees with our data that pneumococcal colonisation in healthy adults is indeed asymptomatic. Higher symptom severity scores were not a predictor for colonisation.

SPN colonisation is more common in children; therefore, a limitation of this work is the lack of generalisability of results to all age groups, however reasonable evidence exists that SPN colonisation in children does cause nasal symptoms[10, 17]. Another limitation is that only one serotype was assessed in this study, SPN 6B. This particular serotype is not thought to be present in the community and therefore it is very unlikely that the participants would have pre-existing immunity from previous exposure. A previous study suggested that the presence of symptoms could be dependent on the serotype of pneumococcus[17]. The authors reported that colonisation with serotype 19F was strongly associated with symptoms such as coryza, sneezing, cough and expectoration. However, these children were recruited from a paediatric hospital emergency room, the study did not report on the diagnosis given to these patients therefore upper or lower respiratory infection may have been the cause of these symptoms rather than solely SPN colonisation[17].

Rodrigues et al found that rhinitis symptoms, rates of colonisation with SPN and *Haemophilus Influenzae (Hi)* in pre-school children decreased with age. Symptoms of rhinitis were reported using the Symptoms of Nasal Outflow Tally (SNOT) score. Both SPN and *Hi*

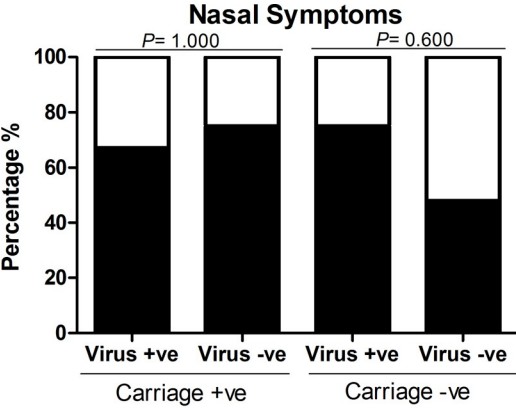

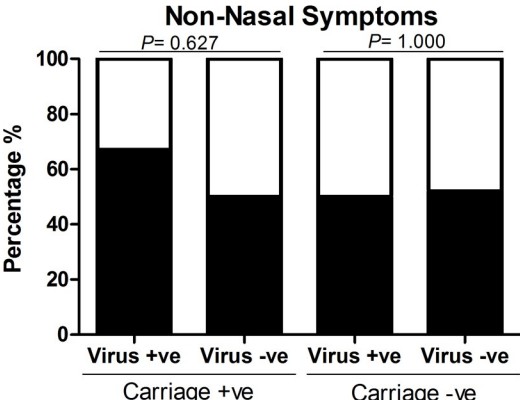

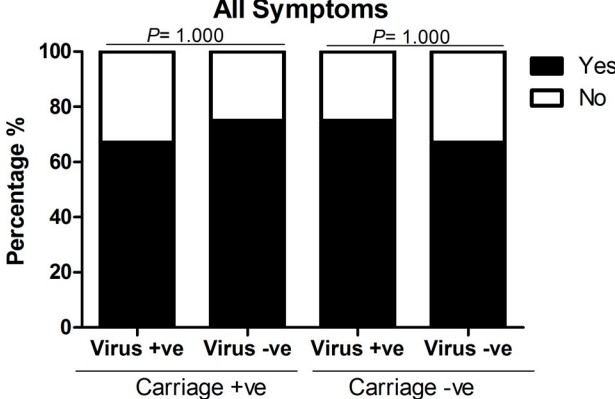

**Fig 6. Comparison of nasal, non-nasal and all symptoms in carriage positive and negative participants with and without viral infection.** Each bar chart shows a comparison of the percentage of carriage positive and negative participants, after inoculation with S. pneumoniae serotype 6B or 23F, who reported symptoms (nasal, non-nasal and all symptoms), between those infected with a virus and those without viral infection. Participants were deemed symptomatic if they scored >2, or >1 point above baseline for any of the five nasal or non-nasal symptoms on the visual analogue scale. The number of participants reporting symptoms between those infected with a virus and those without viral infection were statistically compared using Fishers Exact and deemed significant if P = ≤0.05. There was no significant difference in the number of participants reporting nasal, non-nasal and all symptoms between those with viral infection and those without viral infection in both the carriage positive (nasal symptoms OR 0.67 [95% CI: 0.06–6.88], P = 1.000, non-nasal symptoms OR 2.00 [95% CI: 0.22–17.90] P = 0.627, all symptoms OR 0.67 [95% CI: 0.06–6.88] P = 1.000) and carriage negative groups (nasal symptoms OR 3.23 [0.30–35.13] P = 0.600, non-nasal symptoms OR 0.93 [95% CI: 0.11–7.59] P = 1.000, all symptoms OR 1.50 [95% CI: 0.14–16.55] P = 1.000).

colonisation were strongly associated with increased SNOT scores in children <5 years (p = 0.002 and 0.001) whereas colonisation with *S. aureus* was negatively associated with SNOT scores (p = 0.04). Interestingly, 40% of asymptomatic children (low SNOT score) were in fact SPN colonised. However, when the data was analysed considering age, the association between SPN colonisation and SNOT scores was weak (p = 0.06) whereas the association between SNOT scores and *Hi* colonisation remained strong (p = 0.003). This suggest that *Hi* may stimulate rhinitis in children to increase transmission[9]. The study by Rodrigues et al does not, however, report the effect of co-colonisation (SPN and virus) on symptoms.

Our results suggest that in adults co-colonisation (SPN and virus) is also an asymptomatic process with similar rates of nasal and non-nasal symptoms reported in all groups. Our results did show that asymptomatic viral infection at baseline was associated with the acquisition of SPN colonisation in adults. This is in keeping with results in children which found that asymptomatic viral infections had a large effect on SPN colonisation[18]. They reported that the proportion of children with SPN colonisation was higher during prompted visits for review of upper respiratory tract infections (URTI) symptoms rather than for regular planned study follow up visits. Due to the small sample size of SPN and virus co-colonisers (n = 6), it is difficult to make strong assumptions about the symptomology of co-infection from our study. Viral swabs were also only performed at baseline (up to 7 days prior to inoculation) therefore we cannot assess correlation between symptoms and viral status at each point, nor was density measured.

In conclusion we have provided evidence to support the hypothesis that neither nasopharyngeal inoculation nor experimental pneumococcal colonisation cause nasal or non-nasal symptoms in adults. Our results suggest that asymptomatic viral infection prior to nasopharyngeal inoculation or experimental SPN colonisation does not increase nasal or non-nasal symptoms. A better understanding of the process of viral co-infection in adults and the symptoms caused by viral infection prior to or following acquisition of SPN colonisation is needed and would add to this study's preliminary data. A key question, given the difference between adults and children, is the association between colonisation symptoms and transmission; our study indicates that pneumococcal colonisation in adults is asymptomatic, but does not address transmission dynamics.

## Supporting information

**S1 Fig. TREND checklist.**
(TIF)

**S2 Fig. Experimental human pneumococcal carriage (EHPC) study protocol.**
(PDF)

**S3 Fig. Symptoms study flowchart.**
(DOCX)

## Acknowledgments

We would like to thank the members of our DMSC—Professor Robert C Read, University of Southampton (Chair), Professor David Lalloo and Dr Brian Faragher, Liverpool School of Tropical Medicine (LSTM), the respiratory research nurses, Clinical Research Unit staff, the Infectious diseases team, RLBUHT laboratories and clinical trials pharmacy.

This work would not have been possible without members of the LSTM respiratory team; Debbie Jenkins for data input, Jessica Owugha and Shaun Pennington for their involvement

with statistical analysis and Jenna Gritzfeld for involvement in bacterial inoculum preparation, sample processing, storage and interpretation of samples and results.

We would like to acknowledge the Comprehensive Local Research Network for their support.

## Author Contributions

**Conceptualization:** Stephen B. Gordon, Daniela M. Ferreira, Angela D. Wright, Andrea M. Collins.

**Data curation:** Ashleigh Trimble, Daniela M. Ferreira, Angela D. Wright, Andrea M. Collins.

**Formal analysis:** Ashleigh Trimble, Ryan E. Robinson, Daniella McLenaghan, Duolao Wang, Stephen B. Gordon, Angela D. Wright, Andrea M. Collins.

**Funding acquisition:** Stephen B. Gordon, Daniela M. Ferreira.

**Investigation:** Ashleigh Trimble, Angela D. Wright, Andrea M. Collins.

**Methodology:** Ashleigh Trimble, Duolao Wang, Daniela M. Ferreira, Angela D. Wright, Andrea M. Collins.

**Project administration:** Carole A. Hancock, Angela D. Wright.

**Supervision:** Stephen B. Gordon, Daniela M. Ferreira.

**Writing – original draft:** Ashleigh Trimble, Angela D. Wright, Andrea M. Collins.

**Writing – review & editing:** Ashleigh Trimble, Victoria Connor, Ryan E. Robinson, Daniella McLenaghan, Duolao Wang, Stephen B. Gordon, Daniela M. Ferreira, Angela D. Wright, Andrea M. Collins.

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
