## [Decision Letter · Decision Letter 0]

1 Dec 2019

PONE-D-19-25985

Pneumococcal Colonisation is an Asymptomatic Event in Healthy Adults using an Experimental Human Colonisation Model.

PLOS ONE

Dear Dr Robinson,

Thank you for submitting your manuscript to PLOS ONE. After careful consideration, we feel that it has merit but does not fully meet PLOS ONE’s publication criteria as it currently stands. Therefore, we invite you to submit a revised version of the manuscript that addresses the points raised below during the review process.

We would appreciate receiving your revised manuscript by Dec 30 2019 11:59PM. To enhance the reproducibility of your results, we recommend that if applicable you deposit your laboratory protocols in protocols.io, where a protocol can be assigned its own identifier (DOI) such that it can be cited independently in the future. For instructions see: http://journals.plos.org/plosone/s/submission-guidelines#loc-laboratory-protocols

We look forward to receiving your revised manuscript.

Kind regards,

Ray Borrow, Ph.D., FRCPath

Academic Editor

PLOS ONE

Journal Requirements:

3. Thank you for submitting your clinical trial to PLOS ONE and for providing the name of the registry and the registration number. The information in the registry entry suggests that your trial was registered after patient recruitment began. PLOS ONE strongly encourages authors to register all trials before recruiting the first participant in a study.

1) your reasons for your delay in registering this study (after enrolment of participants started);

2) confirmation that all related trials are registered by stating: “The authors confirm that all ongoing and related trials for this drug/intervention are registered”.

Please also ensure you report the date at which the ethics committee approved the study as well as the complete date range for patient recruitment and follow-up in the Methods section of your manuscript.

4. Please ensure that you refer to Figure 5 in your text as, if accepted, production will need this reference to link the reader to the figure.

Reviewers' comments:

Reviewer's Responses to Questions

**Comments to the Author**

1. Is the manuscript technically sound, and do the data support the conclusions?

Reviewer #1: Yes

Reviewer #2: Yes

2. Has the statistical analysis been performed appropriately and rigorously? 

Reviewer #1: Yes

Reviewer #2: Yes

3. Have the authors made all data underlying the findings in their manuscript fully available?

Reviewer #1: Yes

Reviewer #2: Yes

4. Is the manuscript presented in an intelligible fashion and written in standard English?

Reviewer #1: Yes

Reviewer #2: Yes

5. Review Comments to the Author

Reviewer #1: The manuscript by Trimble and colleagues present additional results from an experimental human challenge model, describing whether experimental pneumococcal colonisation in healthy adults is a symptomatic event. This manuscript is a resubmission from 2016 (but due to time limits is now a new submission) which has addressed some of the queries from the initial review. This study reports some interesting data but is nevertheless limited mainly due to the small sample size, especially for the secondary analyses.

Some points for consideration:

1. The pneumococcal colonisation rate was 30.4% (14/46), which was solely due to serotype 6B, and actually 48% if only those who were inoculated with 6B are included. This seems quite high for an otherwise healthy adult population. The authors state that this model mimics natural pneumococcal exposure but the high carriage rate may suggest otherwise. Can the authors provide some comment on the colonisation rate they observed with respect to what has been reported among UK healthy adults or more generally? Some explanation of this would greatly benefit the manuscript. The authors also argued against providing some comment on the lack of 23F colonisation but I tend to think that this would also be worth noting in the discussion, given that both these are commonly carried serotypes.

2. Figures 2 and 3 could be combined since they are both essentially reporting the same data but with a different comparison. Also, it should be Fisher’s exact test, not Fischer’s exact test.

3. Can the authors comment on the apparent high rate of nasal and non-nasal symptoms in the control group and whether they believe this could have contributed to the lack of any clinical symptoms being demonstrated for the other groups? An independent assessor may have been useful here.

4. There are a couple of instances where what is reported in the results is inconsistent with the data or discussion. For example, on lines 192-193 it states that “This study does not however report the effect of co-colonisation (SPN and virus) on symptoms” when this is clearly shown in Figure 5. Also on lines 207-208, it states that “…experimental SPN colonisation does not increase nasal or non-nasal symptoms” when in the results (lines 156-157) it states that “Experimental SPN colonisation rates were higher in the presence of virus….(p<0.05)”. Please correct. The p-value, if indeed significant, should be included in the Figure.

5. What were the viruses that were detected, and were there any associations between specific viruses?

6. For most of the reported p-values there are no = sign. Please include.

Reviewer #2: General comments

This study uses a human challenge model to examine if symptoms are experienced in the context of pneumococcal colonisation among adults. The structure of the paper is acceptable, however there could be improvement made in the finer details, e.g. error bars/confidence intervals and raw numbers

I think it is also important to mention the limitations of this study e.g. not an RCT, low sample size, and only one serotype successfully colonised. Consequently I think some of the statements in the discussion should have the language softened, as this study is not the definitive study which demonstrates that colonisation is asymptomatic among adults. Rather it might suggest this, but further work using more serotypes, in larger RCTs is needed.

Specific comments

#Abstract

Methods section line 46

" ... reported between groups was compared ..." to groups were compared

Results section line 49

S. pneumoniae (29 with serotype 6B, 17 with serotype 23F ...

Conclusion section line 58

Pneumococcal colonisation using the EHPC was asymptomatic ...

#Introduction

line 69 - correct typo, H. influenzae

line 70-73 - could be better worded

lines 83-87 - suggest rearranging this paragraph e.g.

The experimental pneumococcal challenge model (EHPC) mimics natural colonisation in healthy ....vaccines in randomised control trials. We aimed to use the EHPC to investigate if the process is symptomatic....

#Methods

Was a sample size calculation conducted?

#Results

line 134-137

please provide actual numbers, proportions, confidence intervals, and p-values.

figures - please add in confidence intervals.

#Discussion

Lines 164 and 212

I would tone down the language here e.g.

This study provides evidence supporting the hypothesis that SPN colonisation among adults is asymptomatic

line 171

Other than sample size, key limitations also include: the study wasn't randomised, only one serotype was assessed in this study, and pre-existing immunity.

line 213

I would suggest adding a sentence to say that larger randomised studies using a wider variety of

6. PLOS authors have the option to publish the peer review history of their article (what does this mean?). If published, this will include your full peer review and any attached files.

Reviewer #1: No

Reviewer #2: Yes: Rama Kandasamy

---

## [Author Response · Author response to Decision Letter 0]

6 Jan 2020

Review Comments to the Author

Reviewer #1:

 The manuscript by Trimble and colleagues present additional results from an experimental human challenge model, describing whether experimental pneumococcal colonisation in healthy adults is a symptomatic event. This manuscript is a resubmission from 2016 (but due to time limits is now a new submission) which has addressed some of the queries from the initial review. This study reports some interesting data but is nevertheless limited mainly due to the small sample size, especially for the secondary analyses.

Some points for consideration:

1. The pneumococcal colonisation rate was 30.4% (14/46), which was solely due to serotype 6B, and actually 48% if only those who were inoculated with 6B are included. This seems quite high for an otherwise healthy adult population. The authors state that this model mimics natural pneumococcal exposure but the high carriage rate may suggest otherwise. Can the authors provide some comment on the colonisation rate they observed with respect to what has been reported among UK healthy adults or more generally? Some explanation of this would greatly benefit the manuscript. The authors also argued against providing some comment on the lack of 23F colonisation but I tend to think that this would also be worth noting in the discussion, given that both these are commonly carried serotypes.

Thank you for your feedback, the EHPC model is able to artificially induce the otherwise naturally occurring phenomenon of pneumococcal carriage by pipetting the live bacteria directly into the participant’s nasopharynx. As the EHPC inoculation is more efficient than the ‘natural process’ the rate of colonisation is higher. This has now been added to the discussion. We have added an explanation for the difference in colonisation rate between serotypes as requested. 

2. Figures 2 and 3 could be combined since they are both essentially reporting the same data but with a different comparison. Also, it should be Fisher’s exact test, not Fischer’s exact test.

Thank you, we have now corrected the spelling of Fisher’s exact test. We feel that combining the two figures together may affect the clarity of the underlying message, and therefore have respectfully chosen to keep them separate. We have however added a legend to all figures to make them easier to interpret. 

3. Can the authors comment on the apparent high rate of nasal and non-nasal symptoms in the control group and whether they believe this could have contributed to the lack of any clinical symptoms being demonstrated for the other groups? An independent assessor may have been useful here.

We recognise that there is a high number of symptoms in the control group. We believe this may be due in part to the small study sample size but also reflects the high sensitivity of the Likert scale used to record any symptoms. Since the symptoms were self-reported by participants and they were blinded to their intervention group, we feel that an independent assessor would have been unlikely to improve this. 

4. There are a couple of instances where what is reported in the results is inconsistent with the data or discussion. For example, on lines 192-193 it states that “This study does not however report the effect of co-colonisation (SPN and virus) on symptoms” when this is clearly shown in Figure 5. Also on lines 207-208, it states that “…experimental SPN colonisation does not increase nasal or non-nasal symptoms” when in the results (lines 156-157) it states that “Experimental SPN colonisation rates were higher in the presence of virus….(p<0.05)”. Please correct. The p-value, if indeed significant, should be included in the Figure.

Thank you, the comment regarding symptoms from co-colonisation refers to the Rodrigues at al study. This has been made clearer in the text to avoid confusion.

In the discussion we note that experimental SPN colonisation does not increase nasal or non-nasal symptoms. In the results section we note that experimental SPN colonisation was higher in the presence of virus. This is referring to two separate points. Participants who are positive for the presence of a virus and SPN (‘co-colonised’) did not appear to have greater rates of symptoms. The figure relating to this data (Figure 5) has been edited to include the relevant p- values and a legend has been added for clarity. 

5. What were the viruses that were detected, and were there any associations between specific viruses?

A variety of viruses were detected, however since these were only present in very small numbers, we were unable to ascertain any relationship between symptoms and these specific viruses. In future studies we aim to further investigate this relationship in more detail. 

For most of the reported p-values there are no = sign. Please include.

Thank you, this has now been added.

Reviewer #2: General comments

This study uses a human challenge model to examine if symptoms are experienced in the context of pneumococcal colonisation among adults. The structure of the paper is acceptable, however there could be improvement made in the finer details, e.g. error bars/confidence intervals and raw numbers

I think it is also important to mention the limitations of this study e.g. not an RCT, low sample size, and only one serotype successfully colonised. Consequently I think some of the statements in the discussion should have the language softened, as this study is not the definitive study which demonstrates that colonisation is asymptomatic among adults. Rather it might suggest this, but further work using more serotypes, in larger RCTs is needed.

Thank you, we have modified the manuscript to take this into account. Changes have been made to the abstract, discussion and conclusion to recognise this. 

Specific comments

Abstract

Methods section line 46

" ... reported between groups was compared ..." to groups were compared

Thank you, this has now been updated. 

Results section line 49

S. pneumoniae (29 with serotype 6B, 17 with serotype 23F ...

Thanks, this has been updated

Conclusion section line 58

Pneumococcal colonisation using the EHPC was asymptomatic ...

Thanks, this has also been updated as requested. 

Introduction

line 69 - correct typo, H. influenzae

Thanks, this has now been updated 

line 70-73 - could be better worded

This has been re-worded for clarity. 

lines 83-87 - suggest rearranging this paragraph e.g.

The experimental pneumococcal challenge model (EHPC) mimics natural colonisation in healthy ....vaccines in randomised control trials. We aimed to use the EHPC to investigate if 

the process is symptomatic....

Thank you this has now been updated.

Methods

Was a sample size calculation conducted?

Thank you for this feedback. We have not performed a sample size calculation as this was performed as a pilot study. In a larger follow up study we will ensure that a formal sample size calculation is done. 

Results

line 134-137

please provide actual numbers, proportions, confidence intervals, and p-values.

These have been added where required. 

figures - please add in confidence intervals.

Thank you, these have been included where required. A legend has been added to aid the interpretation. 

Discussion

Lines 164 and 212

I would tone down the language here e.g.

This study provides evidence supporting the hypothesis that SPN colonisation among adults is asymptomatic

Thank you, this has been updated. We have taken this into account and altered the discussion and conclusion to reflect this. 

line 171

Other than sample size, key limitations also include: the study wasn't randomised, only one serotype was assessed in this study, and pre-existing immunity.

The lack of randomisation to the allocated group and the use of a single successful serotype has been added as key limitations. As we used SPN 6B, a serotype that is not otherwise present in the community, it is unlikely that the participants have pre-existing immunity to this serotype. A line has been added to the discussion in order to address this. 

line 213

---

## [Decision Letter · Decision Letter 1]

11 Feb 2020

Pneumococcal Colonisation is an Asymptomatic Event in Healthy Adults using an Experimental Human Colonisation Model.

PONE-D-19-25985R1

Dear Dr. Robinson,

We are pleased to inform you that your manuscript has been judged scientifically suitable for publication and will be formally accepted for publication once it complies with all outstanding technical requirements.

With kind regards,

Ray Borrow, Ph.D., FRCPath

Academic Editor

PLOS ONE

Additional Editor Comments (optional):

Reviewers' comments:

Reviewer's Responses to Questions

**Comments to the Author**

1. If the authors have adequately addressed your comments raised in a previous round of review and you feel that this manuscript is now acceptable for publication, you may indicate that here to bypass the “Comments to the Author” section, enter your conflict of interest statement in the “Confidential to Editor” section, and submit your "Accept" recommendation.

Reviewer #1: All comments have been addressed

2. Is the manuscript technically sound, and do the data support the conclusions?

Reviewer #1: Yes

3. Has the statistical analysis been performed appropriately and rigorously? 

Reviewer #1: Yes

4. Have the authors made all data underlying the findings in their manuscript fully available?

Reviewer #1: Yes

5. Is the manuscript presented in an intelligible fashion and written in standard English?

Reviewer #1: Yes

6. Review Comments to the Author

Reviewer #1: (No Response)

7. PLOS authors have the option to publish the peer review history of their article (what does this mean?). If published, this will include your full peer review and any attached files.

Reviewer #1: No

---

## [Editor Report · Acceptance letter]

18 Feb 2020

PONE-D-19-25985R1 

Pneumococcal Colonisation is an Asymptomatic Event in Healthy Adults using an Experimental Human Colonisation Model. 

Dear Dr. Robinson:

I am pleased to inform you that your manuscript has been deemed suitable for publication in PLOS ONE. Congratulations! Your manuscript is now with our production department. 

With kind regards,

on behalf of

Prof. Ray Borrow 

Academic Editor

PLOS ONE